# The concept of optimal planning of a linearly oriented segment of the 5G network

**Viacheslav Kovtun**[1]*, **Krzysztof Grochla**[1], **Elena Zaitseva**[2], **Vitaly Levashenko**[2]

**1** Internet of Things Group, Institute of Theoretical and Applied Informatics Polish Academy of Sciences, Gliwice, Poland, **2** Department of Informatics, University of Žilina, Žilina, Slovakia

* vkovtun@iitis.pl

**Data Availability Statement:** All relevant data are within the manuscript and its Supporting information files.

**Funding:** This research is part of the project No. 2022/45/P/ST7/03450 co-funded by the National

## Abstract

In the article, the extreme problem of finding the optimal placement plan of 5G base stations at certain points within a linear area of finite length is set. A fundamental feature of the author's formulation of the extreme problem is that it takes into account not only the points of potential placement of base stations but also the possibility of selecting instances of stations to be placed at a specific point from a defined excess set, as well as the aspect of inseparable interaction of placed 5G base stations within the framework of SON. The formulation of this extreme problem is brought to the form of a specific combinatorial model. The article proposes an adapted branch-and-bounds method, which allows the process of synthesis of the architecture of a linearly oriented segment of a 5G network to select the best options for the placement of base stations for further evaluation of the received placement plans in the metric of defined performance indicators. As the final stage of the synthesis of the optimal plan of a linearly oriented wireless network segment based on the sequence of the best placements, it is proposed to expand the parametric space of the design task due to the specific technical parameters characteristic of the 5G platform. The article presents a numerical example of solving an instance of the corresponding extremal problem. It is shown that the presented mathematical apparatus allows for the formation of a set of optimal placements taking into account the size of the non-coverage of the target area. To calculate this characteristic parameter, both exact and two approximate approaches are formalized. The results of the experiment showed that for high-dimensional problems, the approximate approach allows for reducing the computational complexity of implementing the adapted branch-and-bounds method by more than six times, with a slight loss of accuracy of the optimal solution. The structure of the article includes Section 1 (introduction and state-of-the-art), Section 2 (statement of the research, proposed models and methods devoted to the research topic), Section 3 (numerical experiment and analysis of results), and Section 4 (conclusions and further research).

## 1. Introduction and state-of-the-art

The strategy for automating the control processes of wireless communication networks is implemented by 3GPP (the main developer of the Technical Specifications for 5G+ generation

Science Centre and the European Union Framework Programme for Research and Innovation Horizon 2020 under the Marie Skłodowska-Curie grant agreement No. 945339.

**Competing interests:** The authors have declared that no competing interests exist.

mobile communications) in the form of a technological concept for the use of Self-Organizing Networks (SON) [1–5], covering all stages of the life cycle of the corresponding communication networks. As defined in the 3GPP technical specifications, SON is a network used by a dedicated control and operational system to automate the management of a 5G network with minimal human intervention. The implementation of SON algorithms in mobile communication networks of the 5G generation is primarily focused on achieving such goals as [6–8]:

- reduction of capital and operating costs of operators (CAPEX/OPEX) during deployment, operation and optimization of the network;

- increasing spectral efficiency, productivity and capacity, improving radio coverage, and increasing the efficiency of resource use of Next Generation Radio Access Networks (NGRAN).

In the context of its intended purpose, SON is considered a driver technology for the development of 5G networks. However, malfunctioning SON network algorithms (for example, after updating cell coverage areas and when connecting new base stations) can pose a threat to the normal functioning of the wireless network and have a significant negative impact, reducing the quality of services of connected users. Considering that SON algorithms are not standardized by 3GPP technical specifications, research into issues related to achieving the abovementioned intended purpose of SON is relevant.

Let's focus our attention on designing the architecture of a 5G network segment with SON, which is deployed over a linear area of finite length. We chose this topology not by chance because the linear organization of space is typical both for urban infrastructure and for intercity communications (highways, railways). When designing a target linearly oriented segment of a 5G network, it is necessary to solve several interrelated problems, namely [9, 10]:

- conduct an analysis and preliminary study of the network deployment location,

- select technical means and protocols,

- determine the architecture of base station placement and evaluate the performance characteristics of the 5G network segment using mathematical modelling.

An effective way to increase the technical and economic indicators of such a project is to optimize the architecture of the created 5G network segment, namely, solving the problem of selecting the optimal set of stations from a given redundant set and determining their locations along the target linearly oriented aria.

The challenge of determining the optimal locations for 5G base stations can be conceptualized as a network design problem involving relays (NDPR), a subject that has garnered significant attention in recent research [11–19]. Originating from the work of Cabral et al. in [15], this problem aims to minimize costs while ensuring that each source node can reach the sink node through a single route within limited transmission distances. Konak's extension [14] introduces a novel problem formulation incorporating a set covering constraints. This formulation forms the basis for a genetic algorithm featuring a specialized crossover/mutation operator, generating feasible paths for each commodity. The relay locations on these paths are then determined by solving the corresponding set covering problem. In [15], Xiao et al. propose a hybrid approach utilizing variable neighbourhood searches. The variable neighbourhood algorithm identifies routes for each source node, and an implicit enumeration algorithm determines optimal relay locations for a given set of routes. Li et al. [12] put forward an iterative meta-heuristic algorithm based on tabu search, addressing the NDPR problem in two steps. However, these studies primarily focus on transmission distance constraints and do not fully

account for the operational dynamics of wireless sensor networks. In contrast, Bagaa et al. [18] and Li et al. [19] establish models for optimal relay node placement, specifically targeting the minimization of energy consumption in wireless signal transmission. Various works in the literature [20, 21] delve into relay node location problems considering signal coverage constraints. Naveen et al. [21] define signal coverage as a location within the communication range of a node eventually connected, possibly via multiple hops, to a sink node. Existing models studying signal coverage often rely on the ideal 0–1 disk channel model [22] and the latest Signal-to-Noise Ratio (SNR) model [16].

Nevertheless, these investigations predominantly concentrate on ascertaining whether all designated areas can be covered, with limited attention given to service quality and timely reliability [23]. In the realm of wireless networks, the paramount concern is whether user needs can be satisfied within specified timeframes. Hence, it becomes crucial to incorporate timely reliability considerations when modelling the location of base stations. Notably, Nigam et al. [11] and Naveen et al. [21] have delved into the optimal placement of relay nodes in wireless sensor networks, taking delay constraints into account. Their objective is to formulate a multi-hop wireless network that meets transmission path delays, ultimately minimizing the number of required relay base stations. However, these investigations uniformly employ the number of hops as the sole criterion for measuring network transmission delay and reliability. This approach overlooks the intricacies of the actual data transmission processand user requirements. Consequently, the results obtained from these studies may not directly offer actionable insights for communication operators.

One of the major challenges faced by current heuristic-based models in addressing the optimization of 5G base station locations is the accurate simulation of Line-of-Sight (LOS) propagation and the coverage of 5G signals in urban settings. Many existing location optimization models for cellular network planning were originally designed for 2G/3G/4G networks [24]. In urban areas, the penetration loss of 2G/3G/4G signals is considerably lower than that of 5G signals [25]. Consequently, the location optimization models developed for earlier generations of cellular networks did not need to account for the simulation of LOS propagation of signals.

Recent research endeavours have employed Geographic Information Systems (GIS) to facilitate the simulation of Line-of-Sight (LOS) propagation and coverage of 5G signals, offering support for optimization and enabling intuitive visual analyses in the planning of 5G networks. For instance, [26] utilized GIS alongside a computational geometry approach to optimize the locations of below-rooftop wall-mounted 5G base stations in urban outdoor areas, with GIS providing crucial support [27]. However, it is worth noting that the computational geometry approach may not be suitable for global optimization. Conversely, recent studies have introduced spatially implicit heuristic algorithm-based models to aid in 5G cellular network planning, particularly in specialized 5G application scenarios [28]. For example, the deployment of 5G base stations on unmanned aerial vehicles (UAVs) has been explored as an effective means to mitigate the penetration loss of 5G signals in urban areas. Consequently, these optimization models for UAV base station deployment do not involve the simulation of LOS propagation of 5G signals.

The papers [29–32] discuss the strategic planning and implementation of wireless heterogeneous networks for smart metering, employing a cross-layer approach. Our methodology integrates considerations from the network layer, addressing routing and flow demands at each link within the network, while simultaneously accounting for limitations at the physical layer, specifically the capacity constraints of short-range technology in a multi-hop configuration. The proposed model utilizes a column generation approach to address the capacitated multi-commodity flow problem, incorporating wireless link capacities, coverage parameters, and associated costs. Those studies incorporate multi-hop routing of packets within a mesh

network composed of smart meters and concentrators connected to a cellular network through base stations. Each link's traffic is represented in a multigraph with occupation percentages, and a scalable routing tree is constructed on a georeferenced map to simulate a real-world deployment. The outcomes illustrate the performance of the proposed model concerning concentrator traffic load, network coverage, and energy consumption reduction. The findings underscore that the integration of multi-hop short-range technology leads to a cost-effective infrastructure by diminishing the number of smart meters requiring a direct connection to cellular technology.

Unlike the previously mentioned studies, our research focuses on a specific optimization problem formulated in the context of organizing a wireless network based on the 5G platform. The specificity is that the mentioned network is deployed along a linear segment, for example, a road, pipeline, or railway. A special feature of the 5G platform is, among other things, the ability to implement several virtual network segments on the base station hardware using Network Slicing technology. In the model proposed in Section 2.1, it is assumed that the frequency resource of the base stations (the cost of which is known) is divided into two virtual segments, one of which is used to serve subscribers, and the second to ensure the network connectivity (for communication with other base stations). The first segment is characterized by higher throughput and a smaller coverage radius, and the second–vice versa. For the deployed network, parameters such as total cost and total delay are taken into account. Within the target linear segment, the coordinates of the points at which the base station can be located are known. The special same type of base stations (gateways) are located at the ends of the target segment. The optimization problem is that it is necessary to place base stations on the target segment in such a way as to maximize the size of the telecommunications coverage of the segment while fulfilling the requirement for each station to communicate with gateways through a system of hosted stations, as well as fulfilling restrictions on the amount of end-to-end delay and the total cost of deployed stations. Thus, the wireless network structure obtained as a result of solving such an optimization problem will be reliable (due to constant communication between base stations and gateways) and efficient (due to maximizing coverage focused on serving subscribers). In addition, the gateway concept allows the integration of multiple line segments into a single network. This allows us to limit ourselves to the analysis of dozens of points of potential location of base stations in the process of searching for the optimal network configuration. In turn, this makes it possible to solve such an optimization problem using exact methods, among which the most suitable is the branch and bound method (subject to its adaptation to the above-formulated specifics of setting the optimization problem). In addition, the proposed mathematical apparatus takes into account that base stations (not gateways) can be of different types (this allows us to take into account the possibility of upgrading already functioning network architectures). Naturally, the last clarification further complicates the final optimization problem. All of the above stated in the complex is a new research problem, as a result of which it is possible to design linearly organized network segments based on 5G base stations that meet the conditions of high reliability, high quality of service and rational use of funds.

Characteristic of the mentioned studies is the variety of methods used to solve the problems of optimal design of wireless network architectures, which does not eliminate the general drawback: the authors deliberately narrow the range of characteristics of the deployed base stations taken into account (most often, to two parameters–coverage semidiameter and cost). The authors also consider the characteristics of the placed base stations to be unified, ignoring that the actual values of the corresponding characteristics depend on the location of the device. At the same time, the aspect of network connectivity (the continuity of interaction between located base stations within the SON) is not considered. The motive of the authors is clear–

they strive to move from multi-parameter optimization problems to single-parameter ones. This makes it possible to increase the probability of the existence of a feasible solutions region and simplifies the solution process as such, but makes the resulting models extremelyabstracted from the real object.

Note that the final stage of designing a wireless network segment is estimating its performance characteristics to verify the compliance of the values of these assessments with the specified requirements [9, 10]. This estimation most often uses simulation modelling, which is a computationally intensive process. If the network characteristics do not meet the specified requirements, we have to return to the previous stages and repeat the iterative design process with new input data to obtain a new architecture for the wireless network segment.

Next, we will present the main elements that determine the orientation, structure and novelty of the presented research.

The **object** of the research is the process of designing the architecture of a linearly oriented segment of the 5G network in the parametric space of controlled qualitative parameters with constraints.

The research **subject** includes the theory and methods of mathematical modelling, functional analysis, combinatorics and operations research.

The **purpose** of the research is formulated as follows: to create a computationally efficient, holistic concept for finding the optimal 5G base station placement plan at specified points within a linear area of finite length.

The **tasks** of the research will be defined as follows:

- present the research object in the form of an extremal problem on a finite set in the form of a specialized combinatorial model,

- to solve the formulated problem, adopt the branch-and-bounds method, which consists of taking into account the specifics of the object of research, presented in the form of a corresponding mathematical model,

- to formulate variants of the procedure for finding a sequence of optimal placement plans of base 5G stations, taking into account the technological characteristics of the latter ones, which differ in terms of computational complexity,

- implement a numerical experiment, the results of which will justify the effectiveness of the created mathematical apparatus.

Now let's formulate the **main contribution** of the research. In the article, the extreme problem of finding the optimal placement plan of 5G base stations at certain points within a linear area of finite length is set. A fundamental feature of the author's formulation of the extreme problem is that it takes into account not only the points of potential placement of base stations but also the possibility of selecting instances of stations to be placed at a specific point from a defined excess set, as well as the aspect of inseparable interaction of placed 5G base stations within the framework of SON. The formulation of this extreme problem is brought to the form of a specific combinatorial model. The article proposes an adapted branch-and-bounds method, which allows the process of synthesis of the architecture of a linearly oriented segment of a 5G network to select the best options for the placement of base stations for further evaluation of the received plans in the metric of defined performance indicators. As the final stage of the synthesis of the optimal plan of a linearly oriented wireless network segment based on the sequence of the best placements, it is proposed to expand the parametric space of the design task due to the specific technical parameters characteristic of the 5G platform.

## 2. Materials and methods

### 2.1. Statement of the research

Suppose that there is a set of multiband base stations $B = \{b_i\}$, $i = \overline{1, n}$, each element of which is characterized by parameters

$$b_i = \left\{ s_i, \left\{ \vec{S}_{ij} \right\}, t_i, v_i \right\}, j = \overline{1, n}, i \neq j,$$

where $s_i$ is the coverage semidiameter of the service supported by the base station $b_i$, focused on users' tasks maintenance; $\left\{ \vec{S}_{ij} \right\}$ is the coverage semidiameter of the service supported by the base station $b_i$, focused on the connection of the $i$-th base station with the $j$-th base station, $b_i, b_j \in B$; $t_i$ and $v_i$ are the bandwidth and cost of the base station $b_i$, respectively.

The maximum amount of funds that can be spent on the deployment of the network architecture $B$ is represented by the value of the parameter $V$. The maximum permissible delay in the transmission of a data packet between the users of the network $B$ is represented by the value of the parameter $D$.

The investigated network has a linear topology, which is deployed on a segment $\rho = [p_0, p_{m+1}]$ of length $l$, where $p_0$ and $p_{m+1}$ are the starting and ending points of the segment $\rho$, respectively. Within the segment $\rho$, a set of points is defined that are admissible for placing base stations of the network $B$:

$$P = \{ p_k \}, P \in \rho, k = \overline{1, m}.$$

The point $p_k$ corresponds to the coordinate $x_k$ within the segment $\rho$, $\forall k$. The coordinates of the start and end points of the segment $\rho$ are defined as $x_0 = x_0(p_0) = 0$ and $x_{m+1} = x_{m+1}(p_{m+1}) = l$, respectively.

The logic of forming such a network infrastructure assumes that specialized base stations $b_0$ and $b_{n+1}$ are located at points $p_0$ and $p_{m+1}$, which are not included in the set of typical base stations $B$: $\{b_0, b_{n+1}\} \notin B$, and act as gateways. At the same time, the coverage semidiameters of specialized base stations $\{b_0, b_{n+1}\}$ for connection with base stations from the set $B$ are represented by the values of the parameters $S_{0i}$ and $S_{(n+1)i}$, respectively. The costs of specialized base stations $\{b_0, b_{n+1}\}$ are not taken into account when forming the value of the parameter $V$ for an arbitrary implementation of the network $B$.

We need to place base stations from a set of $B$ in admissible points $P$ at the segment $\rho$ of length $l$ in such a way as to maximize the aggregate communication coverage of the latter. With:

- each base station from the set $B$ must constantly maintain communication with gateways $\{b_0, b_{n+1}\}$ (directly or through other base stations of the same type),

- the cost of implementing the found base stations $B$ placement plan should not exceed the value of $V$,

- the maximum allowable delay of data packet transmission between users within the coverage provided by the found base stations $B$ placement plan should not exceed the value of $D$.

Let's generalize the formulated requirements in the form of an extremal problem on a finite set. An acceptable placement plan of base stations from the set $B$ is the following set of pairs

$$A = \{(p_k, b_i)\}, p_k \in P, k \in \left\{ \overline{1, m} \right\},$$

increasing in value of the coordinate $x_k$, for which the following requirements are met:

1. For each pair $(p_k, b_i)$:

   - on the left: either there is a pair of $(p_q, b_j)$ such that $x_k - x_q \leq S_{ij}$ and $x_k - x_q \leq S_{ji}$, or $x_k - x_0 \leq S_{i0}$ and $x_k - x_0 \leq S_{0i}$;

   - on the right: either there is a pair of $(p_g, b_f)$ such that $x_g - x_k \leq S_{if}$ and $x_g - x_k \leq S_{fi}$, or $x_{m+k} - x_k \leq S_{i(n+1)}$ and $x_{m+1} - x_k \leq S_{(n+1)i}$;

2. At each point from the set $P$, no more than one base station from the set $B$ can be located;

3. The inequality $\sum_{i \in B_\delta} \overline{D}_i \leq D$ is satisfied, where $B_\delta$ is the set of deployed base stations from the set $B$, and $\overline{D}_i$ is the average data packet transmission delay determined for the base stations from the set $B_\delta$;

4. The total cost of base stations from the set $B_\delta$ does not exceed the value of $V$.

Note that the fulfilment of requirement 1 guarantees that each base station from the set $B_\delta$ will communicate with the gateways $\{b_0, b_{n+1}\}$ either directly or through other stations of this set.

Let's match each acceptable placement plan $A$ with the corresponding coverage value $y(A)$, which is interpreted as the total coverage for all base stations from the set $B$ included in the placement plan $A$. In this context, we define the concept of non-coverage of a segment $\rho$ as $h(A) = l - y(A)$.

We denote the set of all admissible placement plans $A$ by $W$. Accordingly, let us present the extremal problem described above in the combinatorial form:

$$h(A^*) = \min\{h(A), A \in W\}. \tag{1}$$

Finally, we denote the entire set of options for placing base stations from the set $B$ on the given set $P$ as $\Omega$. Wherein, $B_\delta \in \Omega$.

## 2.2. Representation of the set $\Omega$ in the form of a branching tree

We will focus the material of Section 2 on the formalization of the solution of the extremal problem (1) by the branch-and-bounds method [33]. In this context, it is necessary to determine the procedure for forming a branching tree (binary search tree) for the set $\Omega$ to which the root vertex corresponds. The basis of the procedure we are looking for is the rule of dividing the original set into subsets. Each subset will correspond to a vertex on the branching tree. The tree itself is oriented relative to the root vertex. So, let's determine the rule for dividing the original set into subsets and the procedure for moving along the branching tree.

Let's redefine the set $\Omega$ as $W_{c = 0}$, where the subscript $c$ fixes the iteration number. Starting from $c = 0$, at each $c$-th iteration we will split the original subset $W_c$ into two derived subsets $W_c^1$ and $W_c^2$. The equalities

$$W_c^1 \cup W_c^2 = W_c, \tag{2}$$

$$W_c^1 \cap W_c^2 = \varnothing \tag{3}$$

are fulfilled for derived subsets (look at the Fig 1).

As a rule that determines the division, we introduce a binary value $\beta_{ki}$. Let $\beta_{ki} = 1$, if the base station $b_i \in B$ will be located at the point $p_k$. Accordingly, $\beta_{ki} = 0$ if no base station from the set $B$ is located at point $p_k$. Next, we will associate the partitioning rule $\beta_{ki} = 1$ with a subset of $W_c^1$, and we will associate the partitioning rule $\beta_{ki} = 0$ with a subset of $W_c^2$.

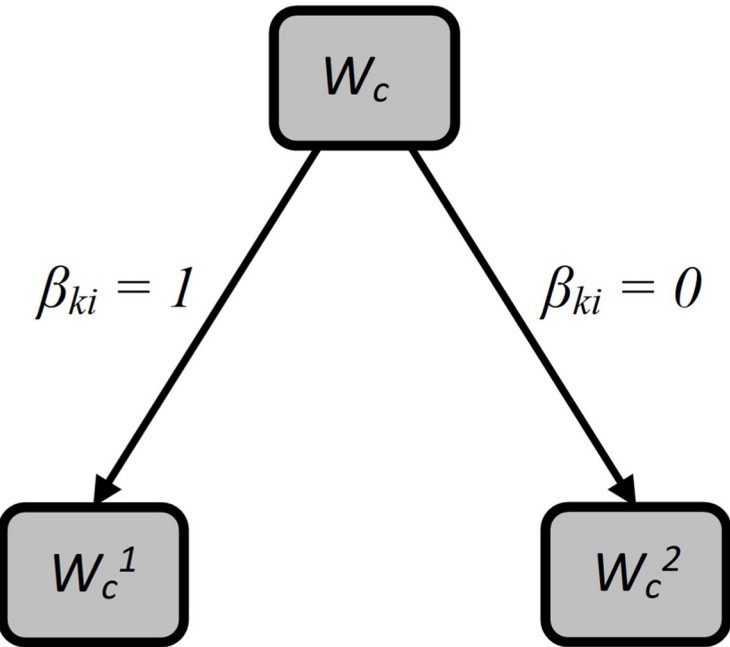

**Fig 1. Branching a binary search tree.**

At the stage of partitioning any initial set $W_c$, the entire set of partitioning rules B = $\{\beta_{ki}\}$ can be divided into three subsets:

- the subset B$^+$, which includes all cases when $\beta_{ki} = 1$,

- the subset B$^-$, which includes all cases when $\beta_{ki} = 0$,

- the subset B$^\sim$, which includes all cases that are not affected by the value of $\beta_{ki}$ at the current $c$-th iteration.

To split the original set $W_c$, at each iteration, a variable from the subset B$^\sim$ with the smallest index $i$ among all variables with the smallest index $k$ is selected. Thus, a free location point $p_k$ with the smallest index $k$ is first selected and the option of placing a base station $b_i \in B$ with the smallest index $i$ at this point is investigated.

After splitting the next source set $W_c$ into derived subsets $W_c^1$ and $W_c^2$ (Fig 2), the latter ones on the branching tree are redefined as $W_{c+1}$ and $W_{c+2}$, respectively. This movement into the depth of the branching tree is typical. At the same time, if an empty set $W_c$ or a set consisting of one element is obtained as a result of the next partition, then such a vertex of the tree is considered closed, and after its analysis, a reverse transition is made from the closed set $W_c$ to one of the previously formed subsets.

To solve the extremal problem (1), we will use the LIFO rule to move through the branching tree. According to this rule, we will implement movement in depth until we find a closed vertex. At the same time, of the two sets $W_c^1$ and $W_c^2$, the set $W_c^1$ will be the first to be examined for the possibility of closing the corresponding vertex. If the found vertex does not turn out to be closed, then movement in depth along the same branch will be implemented from it. Otherwise, a reverse transition will be made: the last formed vertex from the unconsidered ones (the unclosed vertex with the largest sequence number $c$) will be selected to continue the movement. The LIFO branching procedure will end when all vertices of the tree are closed.

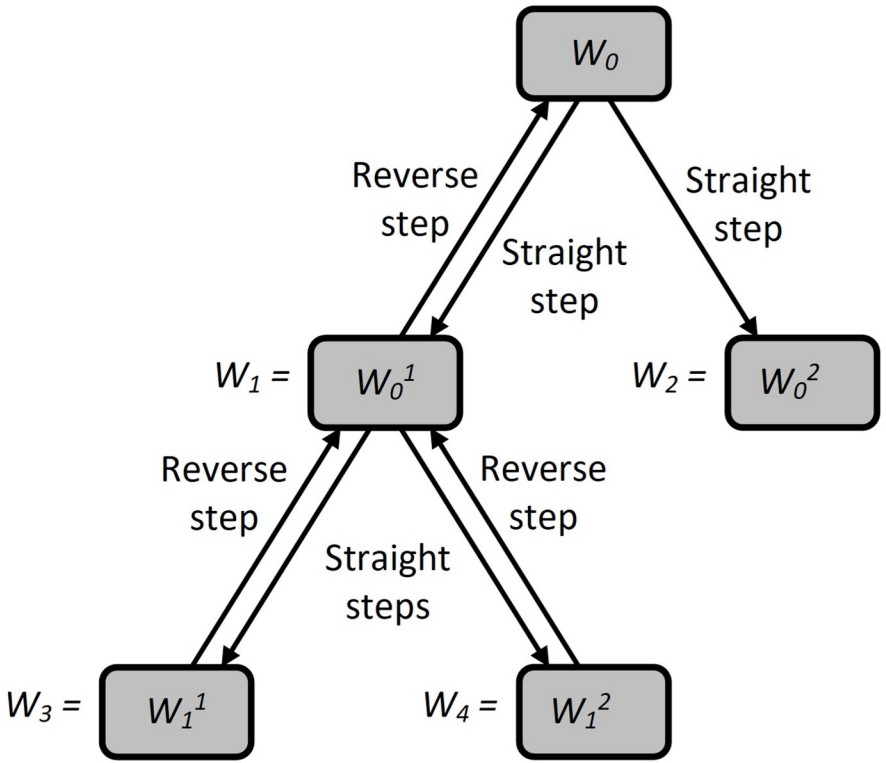

**Fig 2. Moving through the search tree.**

Note that the above-defined vertex selection rule for the implementation of movement in depth gives any set $W_c$ an important property. Suppose that for the studied set $W_c$, $c \in \mathbb{N}$, a base station from the set $B$ has already been placed at the point $p_q$ following the partitioning rule $\beta_{ki}$, which separated the derived set $W_c$ from the original set $W_0$. Then, for all points $p_k$ located to the right of the point $p_q$: $k < q$, the location of base stations from the set $B$ has already been determined (at the same time, some of these points may not contain base stations).

## 2.3. Adaptation of the branch-and-bounds method to the research object

The adaptation of the branch-and-bounds method for solving the extremal problem (1) taking into account the procedure of movement along the branching tree described in Section 2.2 consists in the formalization of the approach to the study of the vertices of the tree for the possibility of their closure.

Let's examine the vertex that corresponds to the set $W_c$. According to the branch-and-bounds method, the vertex can be closed in the following cases:

Case 1. The set $W_c$ is empty, i.e. it is proved that there is no admissible placement plan $A$ in the set $W_c$ with the specified branching rule $\beta_{ki}$;

Case 2. It is proved that the set $W_c$ contains an admissible placement plan $A$, which corresponds to a smaller value of the objective function (1) than the best version of the placement plan $\hat{A}$ among those already found. The value of the non-coverage function $h(\hat{A})$ will be considered extreme, and the placement plan $\hat{A} := A$ will be considered optimal. We consider

the tuple $\langle h(\hat{A}), \hat{A} \rangle$ to be the optimal solution to the problem (1). As the initiating value of the function $h(\hat{A})$, we take the length $l$ of the segment $\rho$;

Case 3. The optimum $\hat{A}$ of the extremal problem (1) on the set $W_c$ is found.

Let's examine these cases in more detail.

Case 1. Checking whether the current derived set $W_c$ is empty is based on requirements 1–4 defined at the end of Section 2.1.

Consider the conditions under which requirement 1 for the set $W_c$, $c \in \mathbb{N}$, is fulfilled. Let the set $W_c$ be formed as a result of splitting the original set using rule $\beta_{qg} = 1$. We check whether each of the semidiameters $S_{gz}$ and $S_{zg}$ is greater than the distance $l_q - l_d$, where $z$ is the index of the base station from the set $B$ located at the point closest to the left $p_d$. If the point closest to the left is $p_0$, then the check is made for semidiameters $S_{q0}$ and $S_{0q}$. If the described checks do not pass, then the set $W_c$ is incompatible, the vertex corresponding to it is considered closed, and the reverse transition is performed according to the rule for the formation of derived subsets defined at the beginning of Section 2.2.

If the set $W_c$ is formed as a result of partitioning the original set $W_0$ using the rule $\beta_{qg} = 0$ and $p_d$ is the point with the largest index among the points from the set $P$, in which the base stations from the set $B$ are already placed (if there are no base stations placed, then $p_d = p_0$), then it should be checked whether among the undistributed base stations from the set $B$ (excluding $b_g$) there is such a base station $b_j$ for which the distance between the points $p_q$ and $p_d$ does not exceed $S_{jz}$ and $S_{zj}$. If the described check is not passed, then the set $W_c$ is incompatible, the vertex corresponding to it is considered closed and the reverse transition is carried out according to the rule for the formation of derived subsets defined at the beginning in Section 2.2.

Requirement 2 is fulfilled as a result of the placement of the next base station from the set $B$. Requirements 3 and 4 are fulfilled as a result of adding the values and delays characteristic of the base stations from the set $B$ located on the segment $\rho$.

Case 2. Let's estimate the amount of non-coverage of the segment $\rho$ of length $l$ for the derived set $W_c$ obtained from the original set when applying the partitioning rule $\beta_{qg} = 1$. For this, we introduce the expression

$$\gamma(q, d, t, g) = \max\left\{ \left( p_q - p_d \right) - \left( s_t + s_g \right), 0 \right\}, \tag{4}$$

where $\gamma$ is the partial non-coverage function, which is defined for any two points $p_q$ and $p_d$ from the set $P$ at $q > d$, in which the base stations $b_t$ and $b_g$ are located, provided that there are no other base stations between these points. For an arbitrary location plan $A$, the non-coverage function $h(A)$ is defined as the sum of the values of functions (4) determined for all base stations located according to the placement plan $A$, including gateways $\{b_0, b_{n+1}\}$.

We define the lower bound of the non-coverage function $h(A)$ for an arbitrary placement plan $A$, which corresponds to the set $W_c$:

$$R(W_c) \leq h(A), A \in W_c. \tag{5}$$

If the inequality $R(W_c) \leq h(\hat{A})$ holds, then the set $W_c$ cannot contain a placement plan $A$ better than the already found $\hat{A}$. Therefore, the corresponding set $W_c$ on the search tree is closed and the reverse transition is performed according to the rule for the formation of derived subsets defined at the beginning in Section 2.2.

In the context of (5), the estimate of the non-covering function for the derived set $W_c$ obtained from the original set when applying the partitioning rule $\beta_{qg} = 1$ is represented by the expression

$$R(W_c) = r_1(W_c) + r_2(W_c), \tag{6}$$

where the term $r_1(W_c)$ is calculated as the sum of the values of functions (4) to the left of the point $p_q$ and the value of the coverage semidiameter of the base station $b_g$ located at this point. The term $r_2(W_c)$ is calculated as the sum of the values of functions (4) to the right of the point $p_q$, that is, on some fragment $\pi$ to the end of the segment $\rho$ (that is, to the point $p_{m+1}$).

We will determine the estimate $r_2(W_c)$ by relaxing the conditions that characterize the possible location of the base stations from the set $B$ on the fragment $\pi$. For this, we need, taking into account the requirements 1–4 defined at the end of Section 2.1, to define a subset $B_\pi \in B$ that contains not yet placed base stations and ensures minimal non-coverage on the fragment $\pi$. We will determine the subset $B\pi$ as a result of solving the corresponding Boolean programming problem. Let's introduce Boolean variables $u_i = \begin{cases} 1 \forall b_i \in B_\mu, \\ 0 \forall \text{else}. \end{cases}$. We formulate the desired Boolean optimization problem in the parametric space of controlled variables $u_i \in \{0, 1\}$ as follows:

$$y = |\mu| - \sum_{u_i \in B_\pi} 2s_i u_i \rightarrow \max, \tag{7}$$

$$\sum_{u_i \in B_\mu} v_i u_i \leq V, \tag{8}$$

$$\sum_{u_i \in B_\mu} u_i \leq n, \tag{9}$$

where $|\pi|$ is the length of the fragment $\pi$ and $n$ is the number of unoccupied points for placing base stations from the set $B$ on the fragment $\pi$. Note that if constraint (8) or constraint (9) is neglected, the optimization problem (7) converges to the integer Knapsack Problem, for the solution of which there is an efficient pseudopolynomial method [34].

The effectiveness of applying assessment in the branch-and-bounds method is determined by its accuracy and the timing of its application. There is no universal solution method for the integer linear programming problem (7). However, based on the above observation, there is a potential opportunity to formalize two less precise estimates, which can be calculated relatively easily. At the same time, the more realistic option is to ignore constraint (9), because, in reality, the number of points for placing base stations significantly exceeds their available number. Next, we will call problem (7) free of constraint (9) reduced. Also, the removal of the restriction on integers inherent in the basic formulation of the problem (7) appears to be potentially promising from the point of view of simplifying the solution process. We will call this version of problem (7) common, because in this interpretation problem (7) becomes an ordinary problem of linear programming for the solution of which there is a universal simplex method. Both the reduced and the usual variation of the formulation of the optimization problem (7) make it possible to find the desired estimates of the term $r_2(W_c)$ from the expression (6) as a result of the solution.

Let's finish the analysis of Case 2 by considering the variant when the derivative set $W_c$ obtained from the original set by applying the partitioning rule $\beta_{qg} = 0$. Obviously, for this variant, the estimate (6) will be equal to the estimate determined for the original set.

Case 3. This case can be implemented only for the set $W_c$ for which there is a single variant of the placement plan $A$. The non-coverage function $h(A)$ for such a placement plan is defined as the sum of the values of all partial non-coverage functions (4) calculated for the points of the segment $\rho$ in which the base stations are located (including gateways $\{b_0, b_{n+1}\}$).

If the requirements 1–4 defined at the end of Section 2.1 are met for the found placement plan $A$ and inequality

$$h(A) < h(\hat{A}), \tag{10}$$

is satisfied, then the value of the function $h(A)$ becomes extreme:$h(\hat{A}) := h(A)$, and the placement plan $A$ becomes a new optimal one:$\hat{A} := A$. Next, the reverse transition is implemented according to the rule for the formation of derived subsets presented at the beginning of Section 2.2. If the inequality (10) is not satisfied, then the values of $h(\hat{A})$, $\hat{A}$ are not updated, but the reverse transition is simply implemented according to the rule for the formation of derived subsets presented at the beginning of Section 2.2.

The operation of the branch-and-bound method ends when all vertices of the search tree are closed. The result of the method is a tuple consisting of $\langle h(A^*) = h(\hat{A}), A^* = \hat{A} \rangle$.

## 2.4. Determination of the optimal architecture and quality characteristics of the elements of the linearly oriented segment of the 5G network

Let us recall the formulation of the extremal problem (1). It is necessary to find such an admissible placement plan $A^*$ of base stations from the set $B$, for which $h(A^*) = \min\{h(A), A \in W\}$.

Let us determine for this problem the sequence of admissible placement plans $\Omega = A^1, A^2, \ldots, A^k$ of the set $W$ for the given $k$. We will form the sequence $\Omega$ so that each placement plan in its composition is no better than the previous one and no worse than the next one: $h(A^1) = h(A^*)$, $h(A^2) = \text{extr}\{h(A), A \in W \backslash A^1\}$, $\ldots$, $h(A^k) = \text{extr}\{h(A), A \in W \backslash A^1 A^2 \cdots A^{k-1}\}$.

Starting with $A^1$, we will sequentially evaluate each subsequent placement plan from the sequence $\Omega$. As soon as we find a placement plan that satisfies all the requirements formulated in Section 2.1, this plan will represent the optimal architecture of the target linear segment of the 5G network. Indeed, for all previous plans, these requirements are not fulfilled, and all subsequent placement plans after the one found in the sequence will not exceed the last one in terms of the value of the criterion $h(A)$.

Let us pay attention to the issue of forming the sequence $\Omega$ using the interpretation of the branch-and-bound method described in Section 2.3. Let us reduce inequality (10) to the form $h(A) \leq h(\hat{A})$. By sequentially arranging all the optimal placement plans found during the method's operation, we will obtain a sequence in which each placement plan will be no worse than the previous one and no better than the next one. To obtain the desired sequence, it is enough to write down the sequence found as a result of the branch-and-bounds method in reverse order.

The advantage of the proposed approach to obtaining the sequence $\Omega$ is simplicity. The disadvantage is that only placement plans no worse than $A^1$ will be included in the resulting sequence. The found sequence $\Omega$ may not contain a placement plan satisfying all optimality criteria. To eliminate this shortcoming, we can resort to the expansion of the set $\Omega$. Let's say that as a result of solving the extremal problem (1), we want to get not only the optimal solution but also all solutions that are worse than the optimal one by a value no greater than $\Delta = l\varepsilon > 0$:

$$h(A) \leq h(\hat{A}) + \Delta, \tag{11}$$

where $\varepsilon$ is the percentage deviation. In practice, the described procedure is implemented by replacing constraint (9) with inequality (11) in the optimization problem (7). By varying the value of $\Delta$, we can get a close to optimal sequence of $\Omega$, devoid of the above-mentioned drawback.

Now let's formalize the calculation of such qualitative characteristics of the obtained optimal architecture of the linearly oriented segment of the 5G network as connectivity semidiameters $S$, coverage semidiameters $s$ and transfer delay between network endpoints (gateways) $D^{p2p}$.

To determine the first two indicators, the primary thing is to establish the energy potential of the network architecture. We characterize this indicator by the equality

$$P_{tran} - L_{rec} + G_{tran} - L_{link} + G_{rec} - L_{rec} = S_{rec} + SFM, \tag{12}$$

where $P_{tran}$ is the transmitter power; $L_{tran}$, $L_{rec}$ are signal losses on the transmitter and receiver sides, respectively; $G_{tran}$, $G_{rec}$ are the main characteristics of the transmitter and the receiver, respectively; $L_{link}$ is a signal loss in the communication channel (air); $S_{rec}$ is the sensitivity of the receiver; $SFM$ is the Signal Fading Margin.

The value of the indicator $L_{link}$ is determined using the Friis formula [33]:

$$L_{link} = 20\lg F_{5G} + 2\lg S_{t-r} - G_{tran} - G_{rec} + \mathsf{K}(F, S_{t-r}), \tag{13}$$

where $F_{5G}$ is the central frequency of the 5G communication system; $S_{t-r}$ is the distance between the transmitter and the receiver; $\mathsf{K}(F, S)$ is a constant, the value of which depends on the units of measurement of the characteristics $F$ and $S$ (for 5G: $F = [GHz]$, $S_{t-r} = [km]$, respectively, $\mathsf{K}(F, S_{t-r}) = 92.45$).

Generalizing the expressions (12), and (13), we will express the indicator $S_{t-r}$. After simplification, we get:

$$S_{t-r} \approx 10\,\widehat{}\,((L_{link} - 20\lg F - \mathsf{K}(F, S_{t-r}))/20). \tag{14}$$

Expression (14) is the basis for calculating semidiameters $s_i$, $S_{ij}$ for base stations from the set $B$.

Now let's formulate an expression for calculating the transfer delay between the endpoints (gateways) of the network. An acceptable level of abstraction will be the simulation of the investigated segment of the 5G network by a queuing system with cross-traffic and $M/M/$1-type nodes. According to Burke's theorem [34], the flow at the exit of the node, which means at the entrance of each subsequent phase, is Poisson. The intensity at the output of each phase is equal to the sum of the intensities of all incoming flows (as noted in Section 2.1, such an intensity is equal to $\mu$). We assume that the real bandwidth of a typical base station from the set $B$ is half of that declared in the specification.

Taking into account the described initial postulates, we characterize the service intensity for the $b_i$-th base station by the expression $\eta_i = 0.5t_i/r$, where $t_i$ is the throughput of the $b_i$-th base station and $r$ is the average length of the data packet. For $\forall b_i \in B$, the load factor is defined as $t_i = n_{is}\mu \sum \mu/\eta_i^2 < 1$, where $n_{is}$ is the number of incoming flows.

Condition $t_i < 1$ is a necessary and sufficient condition for the existence of a stationary mode of operation of the described queuing system. With the specified parameters $t_i$, according to Little's law, the average transfer delay for the $b_i$-th base station of the linearly oriented segment of the 5G network is equal to $\overline{D}_i = t_i/n_{is}\mu(1 - t_i)$. Therefore, the transfer delay

between endpoints (gateways) of the studied network is described by the expression

$$D^{p2p} = \sum_{i=1}^{n} \overline{D}_i. \tag{15}$$

## 3. Results

We will demonstrate the functionality of the mathematical apparatus proposed in Section 2 on an example. Let a linear area $\rho$ with a length of $l$ = 2.3 [km] be specified in the urban development. In this area, six points $|P|$ = 6 suitable for placing 5G base stations, with coordinates $X$ = {0.36, 0.51, 1.15, 1.35, 1.82, 1.91} [km], are selected. A set of typical base stations has a power of $|B|$ = 5. Each element of the set $B$ is characterized in the parametric space $\left\langle G^S_{tran}, L_{t/r}, t, P^S_{tran}, P^S_{rec}, P^s_{rec}, G^s_{rec}, v \right\rangle$, where the parameters $P$ characterize aspects of the power of transmission and reception of information signals (see the description of expression (12)), and the superscript {$S$, $s$} means the orientation of the corresponding parameter, namely:

- the index $S$ means directing the power to ensure the connectivity of base stations in the studied network segment (maintenance of communication between base stations),

- the index $s$ means directing the power to cover the area around the base stations from the set $B$ (supporting communication between users);

- parameter $L_{t/r} = L_{train} = L_{rec}$ characterizes signal loss; parameter $t$ characterizes a throughput of the base station; parameter $v$ characterizes the cost of placing base station $b_i \in B$ at point $p_k \in P$ with coordinates $x_k \in X$, $i = \overline{1, |B|}$, $k = \overline{1, |P|}$.

The values of the parameters $\left\langle G^S_{tran} = 2, L_{t/r} = 1, t = 0.7 \right\rangle$ are the same for $\forall b_i \in B$. The values of the rest of the parameters mentioned above differ for base stations s $b_j, i = \overline{1, |B| = 5}$: $P^S_{tran} = (19, 20, 19, 19, 18)$, $P^S_{rec} = (-77, -77, -77, -75, -77)$, $P^s_{rec} = (-67, -73, -73, -70, -67)$, $G^s_{rec} = (5, 2, 2, 2, 4)$, $v = (4.6, 4.1, 3.8, 4.2, 3.6)$. Parameters $\left\langle G^S_{tran}, L_{t/r}, P^S_{tran}, P^S_{rec}, P^s_{rec}, G^s_{rec} \right\rangle$ are measured in [dB]. The parameters $t$ and $v$ are measured in [Gbps] and [c.u.], respectively.

As mentioned in Section 2.1, specialized base stations (gateways) $\{b_0, b_{|B|+1}\} \notin B$ should be located at the ends of the area $\rho$. Let's assume that these are devices of the same type $b_0, b_6$ with parameters $\left\langle P^S_{tran} = 20, G^S_{tran} = -77, G^S_{rec} = 5, L_{t/r} = 1 \right\rangle$.

FThe maximum amount of funds that can be spent on the deployment of the network architecture $B$ is represented by the value of the parameter $V$ = 12 [c.u.] The maximum permissible delay in the transmission of a data packet between network users is represented by the value of the parameter $D$ = 1.5 [ms]. We assume that the intensity of data packet arrival is $\mu$ = 100 [1/s], and the average length of a data packet is $r$ = 1500 [bytes].

According to expression (14) (taking into account expressions (8), (12), (13)), we calculate the semidiameters $s_i$, $S_{ij}$, $i = \overline{0, 6}$, $j = \overline{0, 6}$, $i \neq j$, for all base stations that should be placed on the area $\rho$ (5 base stations from the set $B$: $b_1, b_2, \ldots, b_5$, and two gateways $b_0, b_6$). When calculating semidiameters $s_i$, the allowance for signal fading is $SFM$ = 14, and when calculating semidiameters $S_{ij}$, we took $SFM$ = 20. For both series of calculations, the centre frequency is $F_{5G}$ = 3.4 [GHz]. The results of the calculations are visualized in the form of diagrams in Figs 3 and 4.

Now we know all the initial data for the application of the adapted branch-and-bound method presented in Section 2.3 for finding the optimal placement plan $A^*$ of base stations

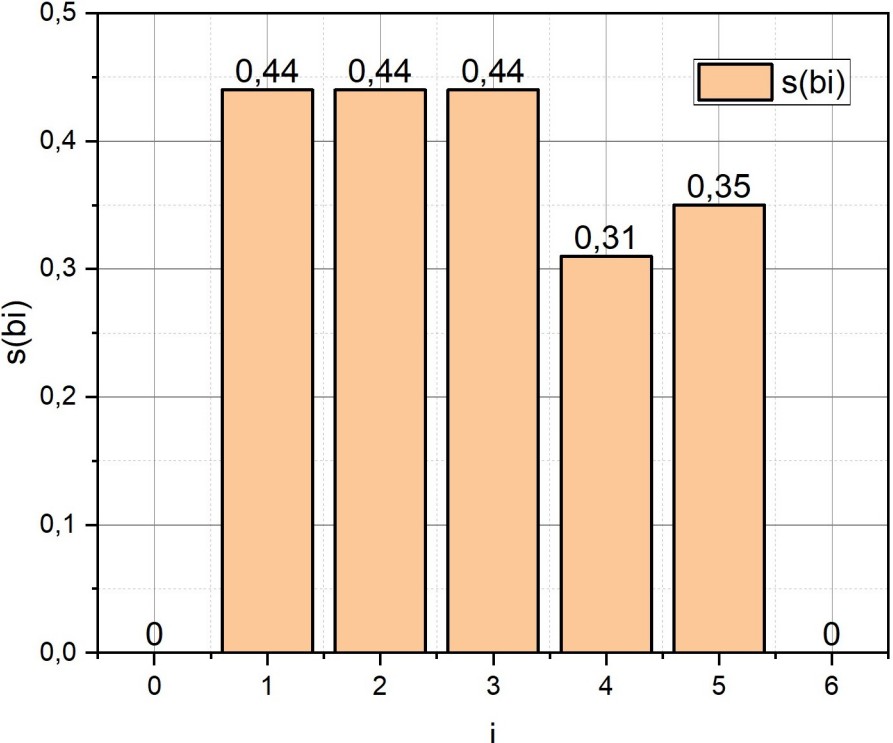

**Fig 3. Results of calculation of coverage semidiameters $s_i$ for base stations $b_i$, $i = \overline{0,6}$.**

from the set $B$ on the aria $\rho$. Recall that the objective function (1) is oriented to the minimization of the non-covering function $h(\hat{A})$.

Fig 5 presents graphs of the dependence of indicators such as:

- the value of the non-coverage function $h(\hat{A})$,

- the cost of implementation of the placement plan $V(\hat{A})$,

- the delay $D(\hat{A})$,

- the number of the vertex of the search tree $N(\hat{A})$,

  on the best placement plan $\hat{A}$ found on the $N_i$-th iteration of the implementation of the adapted branch-and-bounds method.

From Fig 5, we see that the optimal value of the objective function $h(A^*) = 0$ is reached at the $N_i = 9$-th iteration in the 213-th vertex of the search tree, subject to the constraints of $V(A^*) \leq V = 12$, $D(A^*) \leq D = 1.5$. The optimal value of the objective function $h(A^*) = 0$ found on the $N_i = 9$-th iteration means that if the placement plan $A^*$ is implemented in the area $\rho$, there will be no fragments not covered by 5G communication.

Corresponding to the one presented in Fig 5 of the iterative process, the dynamics of the evolution of placement plans $\hat{A} \rightarrow A^*$ are visualized in Fig 6. The optimal placement plan $A^*(P = \{\overline{p_1, p_6}\}) = (b_1, -, b_5, -, -, b_3)$ was found.

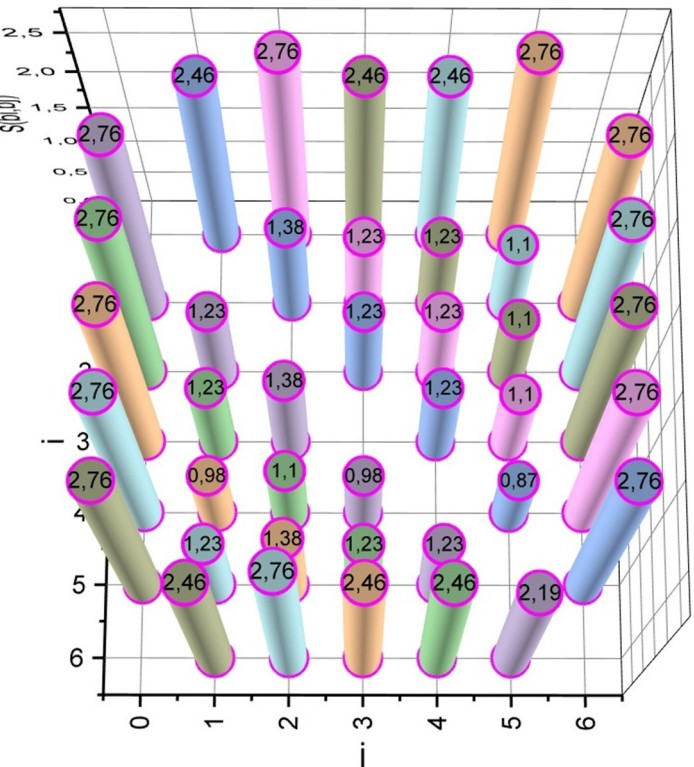

**Fig 4. Calculation results of connectivity semidiameters $S_{ij}$ for base stations $b_i$, $i = \overline{0,6}$, $j = \overline{0,6}$, $i \neq j$.**

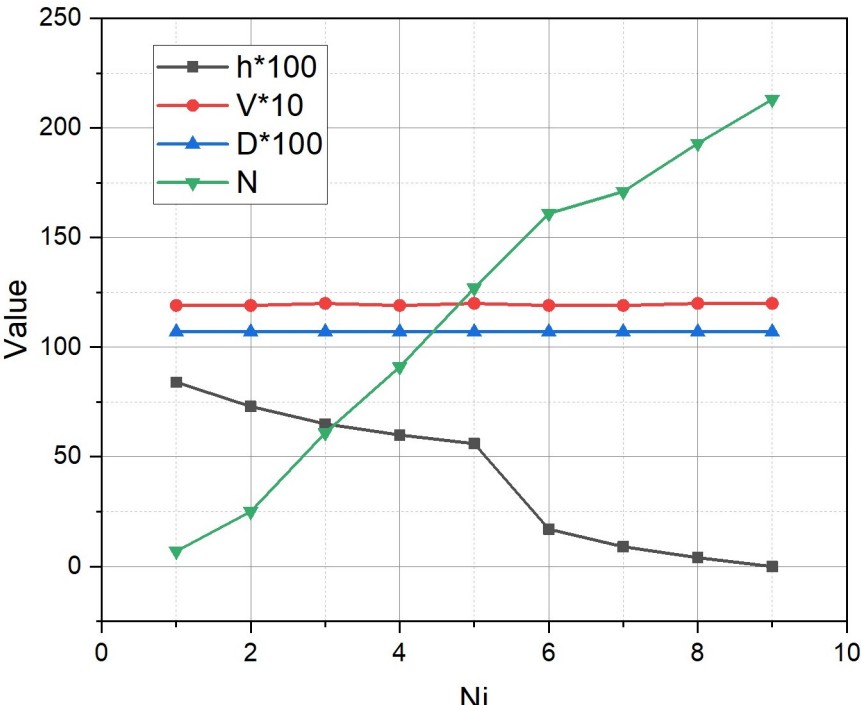

**Fig 5. Dynamics of changes in the values of quality indicators $\langle h, V, D, N \rangle$ depending on the selected placement plan $\hat{A}$ in the process of implementing the adapted branch-and-bounds method.**

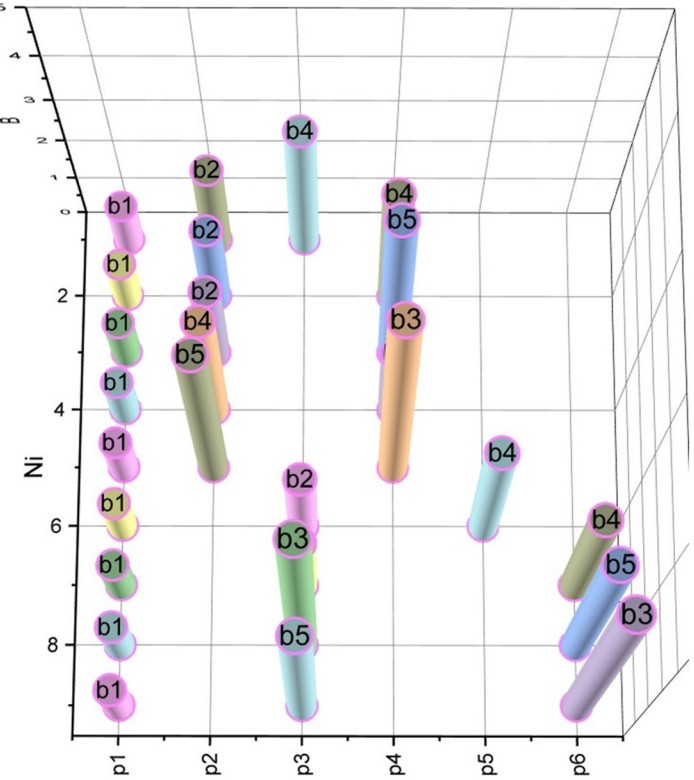

**Fig 6. Corresponds to those presented in Fig 5 results the evolution of $\hat{A} \rightarrow A^*$.**

Finally, let's apply the concept of finding close-to-optimal placement plans reduced to expression (11) (see Section 2.4). Let's enter the threshold deviation value $\varepsilon = 0.5\%$, which corresponds to $\Delta = \varepsilon(l) = 0.5\%(l = 2.3) = 0.0115$ [km]. That is, we accept non-coverage of $h = 11.5$ [m] on the target area $\rho$ with a length of $l = 2.3$ [km] as permissible. Under such assumptions, we found seven approximate placement plans, detailed information about which is visualized in Fig 7.

For all presented in Fig 7 close-to-optimal placement plans have the same delay value $D = 1.07$, but the cost of their implementation differs: $V\left(A_i^{\sim*}, i = \overline{1,7}\right) = \{11.5, 12, 11.5, 12, 11.5, 12, 11.5\}$. Therefore, the placement plans $A_{N_A}^{\sim*}$, $N_A = \{1, 3, 5, 7\}$, assume non-coverage of $h = 11.5$ [m] on the target area $\rho$ with a length of $l = 2.3$ [km] but have a realization cost of 0.5 [c.u.] less than the cost of the optimal placement plan, for which $V(A^*) = 12$.

## 4. Discussion

The results of solving the problem of optimal planning of a linearly oriented segment of the urban 5G network with the main output parameters $l = 2.3$, $|P| = 5$, $|B| = 6$ presented in the previous section testify to the functionality of the mathematical apparatus proposed in Section 2. Let's pay attention to those shown in Fig 5 graphs. The linear nature of the dependencies $D = f(N_i)$ and $V = f(N_i)$ indicates that the work of the method began with a near-optimal superposition of the values of the controlled parameters. The close-to-linear nature of the dependences of $h = f(N_i)$ and $N = f(N_i)$ is a sign of the computational efficiency of the adaptation of the

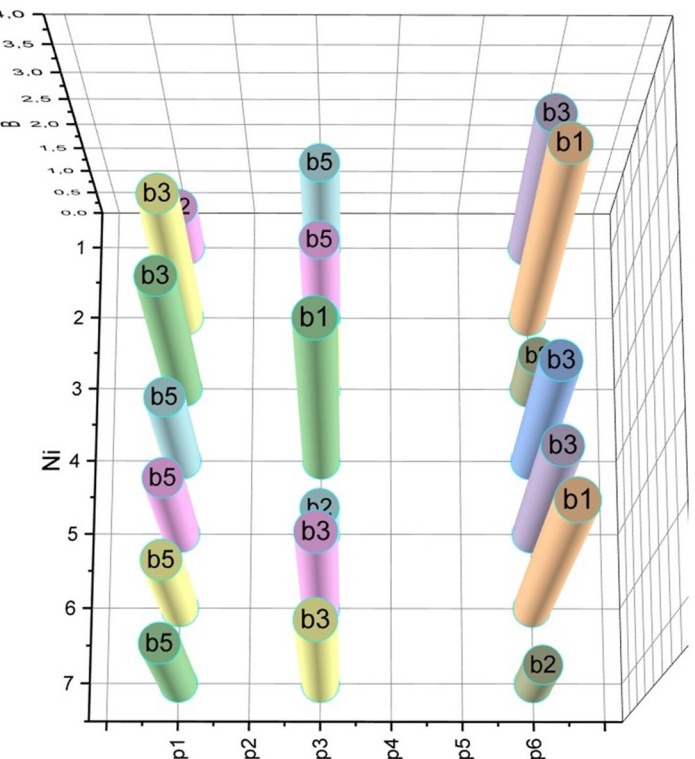

**Fig 7. Placement plans of base stations on aria $\rho$, approximated to the optimum $A^*$ by the value $\varepsilon = 0.5\%$.**

branch-and-bound method proposed in Section 2.3 for solving the extremal problem formulated in Section 2.1. However, the last conclusion needs to be clarified, because in Section 3, an extremal problem of type (1) with a relatively small dimension was solved: $(|P|, |B|) = (5, 6)$. Let's try to clarify the nature of the dependences of $h = f(N_i)$ and s $N = f(N_i)$ for larger-scale extremal problems of type (1).

We will investigate the dependence of such qualitative indicators as the solution time $T$ [s], non-coverage $h$ [m] and solution accuracy $Acc$ [%], for the three options for calculating the term $r_2(W_c)$ from expression (6) described in Section 2.3, which determines the evaluation of the not-covering function $h(A)$ when solving the extremal problem (1) by the adapted branch-and-bounds method. For the definition $r_2(W_c)$, we proposed three variants of the formulation of the optimization problem (7):

- base (*BOP*, optimization problem (7) with constraints (8), (9): integer Linear Programming Problem),

- reduced (*ROP*, optimization problem (7) with constraint (8): integer Knapsack Problem),

- common (COP, optimization problem (7) with constraints (8), (9): ordinary Linear Programming Problem).

Among other initial parameters, note $l(\rho) = 50$ [km], $|P| \in [5, 30]$, $|B_\pi| \in [6, 100]$, $V = 600$ [c.u.] The rest of the parameters are identical to those entered in Section 3.

The calculated dependences of $T = f((|P|, |B_\pi|), \{BOP, ROP, COP\})$, $h = f((|P|, |B_\pi|), \{BOP, ROP, COP\})$, $Acc = f((|P|, |B_\pi|), \{ROP, COP\})$ in the form of diagrams with accumulation are presented in Figs 8–10, respectively, where $(|P|, |B_\mu|)$ are selected sets of values of the

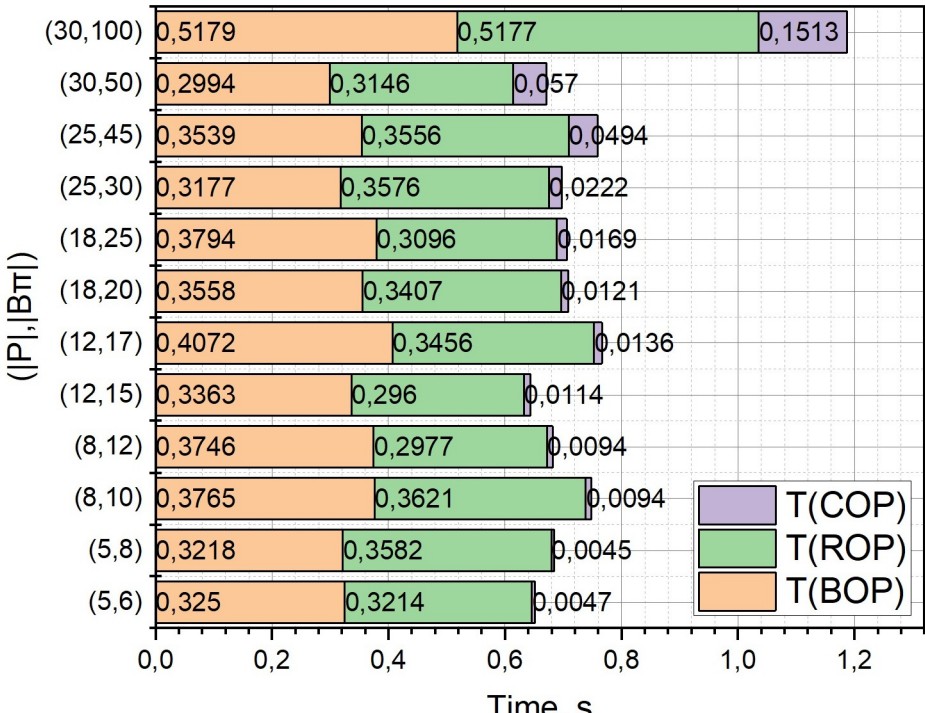

**Fig 8. Diagrams with accumulation, which correspond to the dependences of $T = f((|P|, |B_\pi|), \{BOP, ROP, COP\})$.**

corresponding initial parameters. Note that it is not necessary to calculate the dependence $Acc = f((|P|, |B_\pi|), BOP)$, because the variant of the formulation of the optimization problem $BOP$ is exact.

From those presented in Figs 6–8 results, it can be concluded that to estimate the term $r_2(W_c)$ from the expression (6) with a large dimension of the optimization problem (1), the statement $COP$ should be used. At the same time, we will play slightly lower in accuracy (see Fig 3), but we will significantly (more than 6 times for large pairs $(|P|, |B_\mu|)$) win in the speed of obtaining a solution.

## 5. Conclusions

The need for optimal planning of linearly oriented segments of urban 5G networks is urgent because the landscape of a modern metropolis is organized, for the most part, in a straight line. Separately, we note that 5G base stations are quite compact and do not require separate towers to be erected for installation. Even outside the city, straight lines (highways, railways) constantly accompany a person. Note that such an optimization task can be classified as multi-criteria in the metric of service quality indicators for wireless users.

As is well known, the disadvantage of multi-criteria optimization problems is the probable lack of an analytical solution, and most solution methods are not focused on evaluating suboptimal results, which are approximate, but partially do not correspond to the imposed initial constraints. So, in the article, the extreme problem of finding the optimal placement plan of 5G base stations at certain points within a linear section of finite length is set. A fundamental feature of the author's formulation of the extreme problem is that it takes into account not only the points of potential placement of base stations but also the possibility of selecting

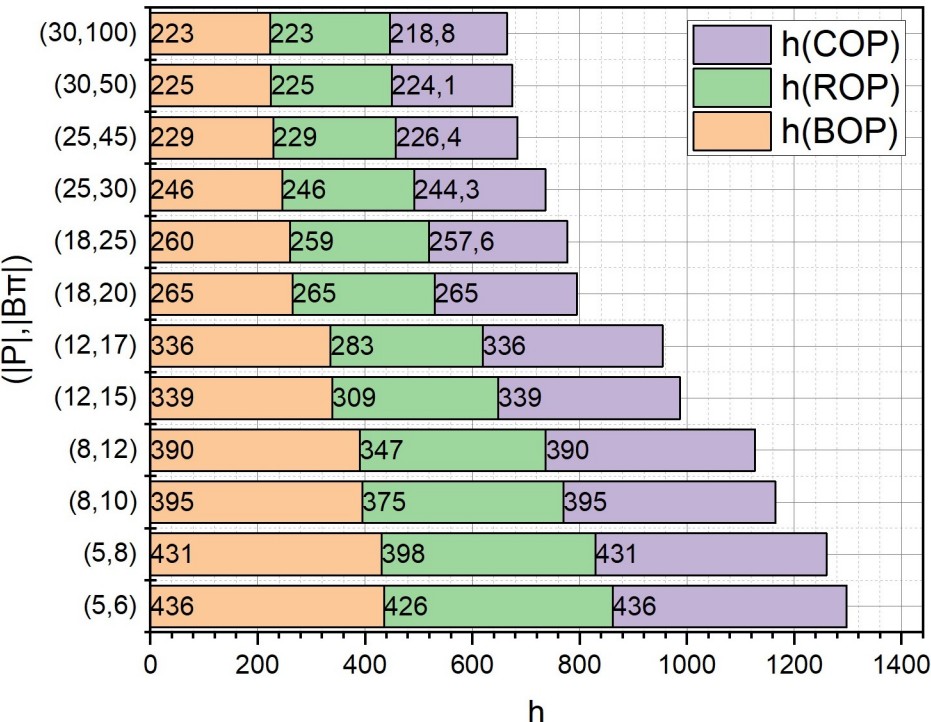

**Fig 9. Diagrams with accumulation, which correspond to the dependences of $h = f((|P|, |B_\pi|), \{BOP, ROP, COP\})$.**

instances of stations to be placed at a specific point from a defined excess set, as well as the aspect of inseparable interaction of placed 5G base stations within the framework of SON. This approach is more realistic, however, it is much more difficult to obtain numerical results. This approach is more realistic, however, it is much more difficult to obtain numerical results.

To overcome these difficulties, the article proposes a method that allows selecting the best options for the placement of base stations in the process of synthesizing the architecture of a linearly oriented segment of a 5G network for further evaluation of the received plans in the metric of defined performance indicators. The mentioned procedure for obtaining a sequence of the best placements is formalized in the form of an adapted branch-and-bounds method. As the final stage of the synthesis of the optimal plan of a linearly oriented 5G network segment based on the sequence of the best placements, it is proposed to expand the parametric space of the design task due to the specific technical parameters characteristic of the 5G platform.

The article presents a numerical example of the synthesis of the optimal placement plan of 5G base stations at a set of admissible points within the target linearly oriented aria of a given length. It is shown that the presented mathematical apparatus allows for the formation of a set of optimal placements taking into account the length of the non-coverage of the target area. To calculate this characteristic parameter, both exact and two approximate approaches are formalized. The results of the experiment showed that for high-dimensional problems, the approximate approach allows for a reduction in the computational complexity of implementing the adapted branch-and-bounds method by more than six times, with a slight loss of accuracy of the optimal solution.

The authors plan to direct **further research** to the expansion of the parametric space of the formulated extreme problem with the nomenclature of characteristic parameters inherent in

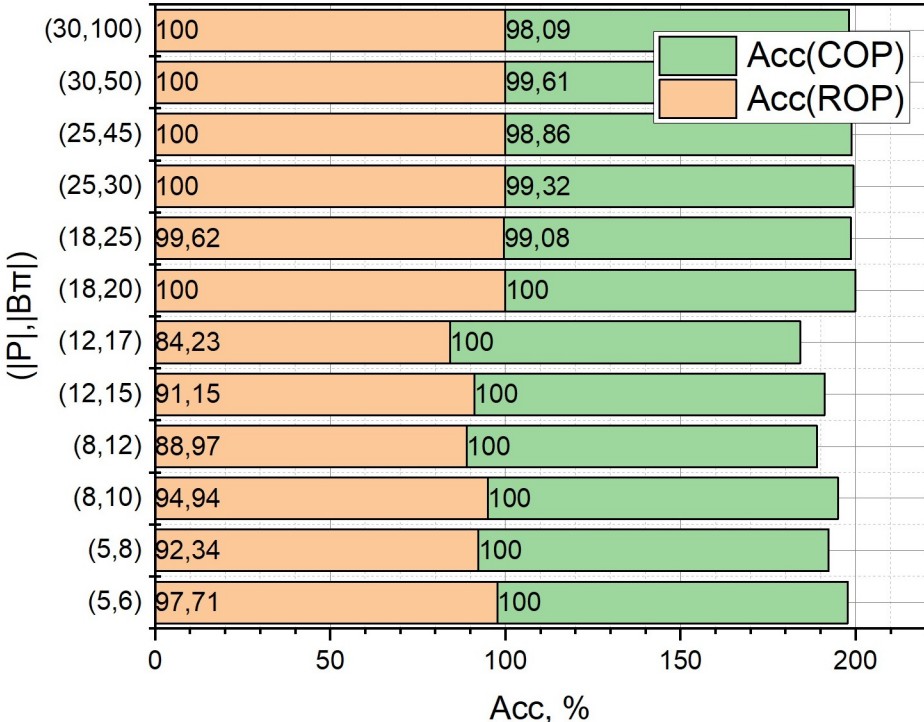

**Fig 10. Diagrams with accumulation, which correspond to the dependences of $Acc = f(|P|, |B_\pi|)$, $\{ROP, COP\}$).**

the 5G platform. This will complicate the solution process, however, it will allow us to take into account the functioning of such relevant 5G technologies as OFDM, Network Slicing, etc.

## Supporting information

**S1 File. Nomenclature.**
(DOCX)

**S1 Data.**
(XLS)

## Acknowledgments

The authors are grateful to all persons and organizations that contributed to the publication of the article.

## Author Contributions

**Conceptualization:** Viacheslav Kovtun.

**Data curation:** Krzysztof Grochla, Elena Zaitseva, Vitaly Levashenko.

**Formal analysis:** Viacheslav Kovtun.

**Funding acquisition:** Viacheslav Kovtun.

**Investigation:** Viacheslav Kovtun.

**Methodology:** Viacheslav Kovtun.

**Project administration:** Viacheslav Kovtun.

**Resources:** Krzysztof Grochla, Elena Zaitseva, Vitaly Levashenko.

**Software:** Viacheslav Kovtun.

**Supervision:** Viacheslav Kovtun.

**Validation:** Krzysztof Grochla, Elena Zaitseva, Vitaly Levashenko.

**Visualization:** Viacheslav Kovtun.

**Writing – original draft:** Viacheslav Kovtun.

**Writing – review & editing:** Viacheslav Kovtun.

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
