## [Decision Letter · Decision Letter 0]

11 Dec 2023

PONE-D-23-33813The concept of optimal planning of a linearly oriented segment of the 5G networkPLOS ONE

Dear Dr. Kovtun,

Thank you for submitting your manuscript to PLOS ONE. After careful consideration, we feel that it has merit but does not fully meet PLOS ONE’s publication criteria as it currently stands. Therefore, we invite you to submit a revised version of the manuscript that addresses the points raised during the review process. Please submit your revised manuscript by Jan 25 2024 11:59PM. If you will need more time than this to complete your revisions, please reply to this message or contact the journal office at plosone@plos.org. Please include the following items when submitting your revised manuscript:A rebuttal letter that responds to each point raised by the academic editor and reviewer(s). You should upload this letter as a separate file labeled 'Response to Reviewers'.A marked-up copy of your manuscript that highlights changes made to the original version. You should upload this as a separate file labeled 'Revised Manuscript with Track Changes'.An unmarked version of your revised paper without tracked changes. You should upload this as a separate file labeled 'Manuscript'.

We look forward to receiving your revised manuscript.

Kind regards,

Mohammed Balfaqih

Academic Editor

PLOS ONE

Journal Requirements:

Reviewers' comments:

Reviewer's Responses to Questions

**Comments to the Author**

1. Is the manuscript technically sound, and do the data support the conclusions?

Reviewer #1: Yes

Reviewer #2: Partly

2. Has the statistical analysis been performed appropriately and rigorously? 

Reviewer #1: No

Reviewer #2: No

3. Have the authors made all data underlying the findings in their manuscript fully available?

Reviewer #1: Yes

Reviewer #2: Yes

4. Is the manuscript presented in an intelligible fashion and written in standard English?

Reviewer #1: Yes

Reviewer #2: Yes

5. Review Comments to the Author

Reviewer #1: 1. Provide the most important numerical experimental results at the end of the abstract.

2. Provide paper structure at the end of the abstract.

3. Literature survey should be expanded with practical and successful solutions to 5G optimization problems.

4. I would recommend writting equations in new line (not inside the text) and numbering them properly (Section 2.1).

5. Also, make sure that each parameter in every equation has been explained in the text.

6. What are the limitations of the proposed approach?

7. If possible, provide comparisons to other available approaches.

8. Avoid using we/our throughout the paper.

Reviewer #2: This paper highlights the need for an optimal planning model for the location of base stations; however, no novelty or innovation is observed compared to recent publications on georeferenced scenarios.

It is suggested that the authors review planning works in georeferenced scenarios and expose the present research's added value.

On the other hand, it is suggested that the authors order the document.

1.- Introduction (Generalities of the problem - 15 references between 2024-2020), 2.- Related Works (Specific works with punctual solutions, summary table of state of the art of other proposals against the current work - 15 references between 2024-2020), 3.- Problem Formulation and Methodology (Table of variables of the mathematical model, mathematical model, pseudocode, methodology flowchart, Big Oh Notation of the algorithm - Forecasting), 4. - Analysis of Results (Metrics performed in Matlab in PDF format or directly from Overleaf), 5.- Conclusions (Direct relation between the objective stated in the abstract vs. the metrics found), 6.-References 2024-2020 (All documents with DOI from ScienceDirect, MDPI, Wiley, PLOS, Taylor & Francis, Springer, Hindawi, IEEE Xplore [transactions, magazines, journals]).

6. PLOS authors have the option to publish the peer review history of their article (what does this mean?). If published, this will include your full peer review and any attached files.

Reviewer #1: No

Reviewer #2: No

---

## [Author Response · Author response to Decision Letter 0]

13 Dec 2023

Dear Editor,

thank you for your time spent for this article.

Due to the fact that the responses to the reviewers were quite detailed, I included them in the submission as a separate file.

Sincerely, Viacheslav Kovtun.

---

## [Decision Letter · Decision Letter 1]

26 Dec 2023

PONE-D-23-33813R1The concept of optimal planning of a linearly oriented segment of the 5G networkPLOS ONE

Dear Dr. Kovtun,

Thank you for submitting your manuscript to PLOS ONE. After careful consideration, we feel that it has merit but does not fully meet PLOS ONE’s publication criteria as it currently stands. Therefore, we invite you to submit a revised version of the manuscript that addresses the points raised during the review process. Please submit your revised manuscript by Feb 09 2024 11:59PM. If you will need more time than this to complete your revisions, please reply to this message or contact the journal office at plosone@plos.org. Please include the following items when submitting your revised manuscript:A rebuttal letter that responds to each point raised by the academic editor and reviewer(s). You should upload this letter as a separate file labeled 'Response to Reviewers'.A marked-up copy of your manuscript that highlights changes made to the original version. You should upload this as a separate file labeled 'Revised Manuscript with Track Changes'.An unmarked version of your revised paper without tracked changes. You should upload this as a separate file labeled 'Manuscript'.If applicable, we recommend that you deposit your laboratory protocols in protocols.io to enhance the reproducibility of your results. Protocols.io assigns your protocol its own identifier (DOI) so that it can be cited independently in the future. For instructions see: https://journals.plos.org/plosone/s/submission-guidelines#loc-laboratory-protocols. Additionally, PLOS ONE offers an option for publishing peer-reviewed Lab Protocol articles, which describe protocols hosted on protocols.io. Read more information on sharing protocols at https://plos.org/protocols?utm_medium=editorial-email&utm_source=authorletters&utm_campaign=protocols.

We look forward to receiving your revised manuscript.

Kind regards,

Mohammed Balfaqih

Academic Editor

PLOS ONE

Journal Requirements:

Reviewers' comments:

Reviewer's Responses to Questions

**Comments to the Author**

1. If the authors have adequately addressed your comments raised in a previous round of review and you feel that this manuscript is now acceptable for publication, you may indicate that here to bypass the “Comments to the Author” section, enter your conflict of interest statement in the “Confidential to Editor” section, and submit your "Accept" recommendation.

Reviewer #1: All comments have been addressed

Reviewer #2: (No Response)

2. Is the manuscript technically sound, and do the data support the conclusions?

Reviewer #1: (No Response)

Reviewer #2: Partly

3. Has the statistical analysis been performed appropriately and rigorously? 

Reviewer #1: (No Response)

Reviewer #2: No

4. Have the authors made all data underlying the findings in their manuscript fully available?

Reviewer #1: (No Response)

Reviewer #2: No

5. Is the manuscript presented in an intelligible fashion and written in standard English?

Reviewer #1: (No Response)

Reviewer #2: Yes

6. Review Comments to the Author

Reviewer #1: (No Response)

Reviewer #2: An optimal wireless network planning model requires evidence of input and innovation on geo-referenced scenarios.

Innovation is not evident in recent publications, as expressed in

https://ieeexplore.ieee.org/abstract/document/10233338

https://www.mdpi.com/1996-1073/13/1/97

On the other hand, it is suggested that the authors order the document.

1.- Introduction (Generalities of the problem - 15 references between 2024-2020), 2.- Related Works (Specific works with punctual solutions, summary table of state of the art of other proposals against the current work - 15 references between 2024-2020), 3.- Problem Formulation and Methodology (Table of variables of the mathematical model, mathematical model, pseudocode, methodology flowchart, Big Oh Notation of the algorithm - Forecasting), 4. - Analysis of Results (Metrics performed in Matlab in PDF format or directly from Overleaf), 5.- Conclusions (Direct relation between the objective stated in the abstract vs. the metrics found), 6.-References 2024-2020 (All documents with DOI from ScienceDirect, MDPI, Wiley, PLOS, Taylor & Francis, Springer, Hindawi, IEEE Xplore [transactions, magazines, journals]).

7. PLOS authors have the option to publish the peer review history of their article (what does this mean?). If published, this will include your full peer review and any attached files.

Reviewer #1: No

Reviewer #2: No

---

## [Author Response · Author response to Decision Letter 1]

28 Dec 2023

Responses to the comments of respected Reviewers are attached to the submission as a separate file.

---

## [Decision Letter · Decision Letter 2]

5 Feb 2024

The concept of optimal planning of a linearly oriented segment of the 5G network

PONE-D-23-33813R2

Dear Dr. Kovtun,

We’re pleased to inform you that your manuscript has been judged scientifically suitable for publication and will be formally accepted for publication once it meets all outstanding technical requirements.

Kind regards,

Mohammed Balfaqih

Academic Editor

PLOS ONE

Additional Editor Comments (optional):

Reviewers' comments:

Reviewer's Responses to Questions

**Comments to the Author**

1. If the authors have adequately addressed your comments raised in a previous round of review and you feel that this manuscript is now acceptable for publication, you may indicate that here to bypass the “Comments to the Author” section, enter your conflict of interest statement in the “Confidential to Editor” section, and submit your "Accept" recommendation.

Reviewer #2: All comments have been addressed

2. Is the manuscript technically sound, and do the data support the conclusions?

Reviewer #2: Yes

3. Has the statistical analysis been performed appropriately and rigorously? 

Reviewer #2: Yes

4. Have the authors made all data underlying the findings in their manuscript fully available?

Reviewer #2: Yes

5. Is the manuscript presented in an intelligible fashion and written in standard English?

Reviewer #2: Yes

6. Review Comments to the Author

Reviewer #2: The authors have made considerable changes.

However, one last recommendation to facilitate the reading of the document is to place the terminology in a table before the mathematical model so that the reader will find it easy to read.

7. PLOS authors have the option to publish the peer review history of their article (what does this mean?). If published, this will include your full peer review and any attached files.

Reviewer #2: No

---

## [Editor Report · Acceptance letter]

4 Apr 2024

PONE-D-23-33813R2 

PLOS ONE

Dear Dr. Kovtun, 

I'm pleased to inform you that your manuscript has been deemed suitable for publication in PLOS ONE. Congratulations! Your manuscript is now being handed over to our production team.

Kind regards, 

on behalf of

Dr. Mohammed Balfaqih 

Academic Editor

PLOS ONE